# Hexagonal Grid-Based Framework for Mobile Robot Navigation

Piotr Duszak [1,*], Barbara Siemiątkowska [1] and Rafał Więckowski [2]

1 Institute of Automatic Control and Robotics, Warsaw University of Technology, 02-525 Warsaw, Poland; barbara.siemiatkowska@pw.edu.pl
2 Łukasiewicz Research Network—Industrial Research Institute for Automation and Measurements PIAP, 02-486 Warsaw, Poland; rafal.wieckowski@piap.lukasiewicz.gov.pl
* Correspondence: piotr.duszak@pw.edu.pl

**Abstract:** The paper addresses the problem of mobile robots' navigation using a hexagonal lattice. We carried out experiments in which we used a vehicle equipped with a set of sensors. Based on the data, a traversable map was created. The experimental results proved that hexagonal maps of an environment can be easily built based on sensor readings. The path planning method has many advantages: the situation in which obstacles surround the position of the robot or the target is easily detected, and we can influence the properties of the path, e.g., the distance from obstacles or the type of surface can be taken into account. A path can be smoothed more easily than with a rectangular grid.

**Keywords:** mapping; data fusion; path planning; hexagonal grid

## 1. Introduction

Over the last few decades, we have observed a rapid development of mobile robotics. Robots such as autonomous vacuum cleaners and lawnmowers are used as standard equipment. The market for autonomous cars and transport within industrial plants is developing.

In 2020, DARPA launched a program called Racer (Robotic Autonomy in Complex Environments with Resiliency) (https://www.darpa.mil/news-events/2020-10-07 accessed on 15 October 2021).

The RACER program aims to develop universal solutions designed to work with various UGV platforms in challenging terrain, taking into account the terrain conditions, at least in terms of adjusting the UGV speed to the environmental conditions. The program aims to develop an autonomy system that will not limit the UGV platform while driving off-road. In other words, if the UGV platform has been designed to drive in rough terrain up to 70 km/h, the autonomy system is to be able to safely navigate the vehicle up to this speed. The program's primary objective is to develop an algorithm that adjusts the local path of the UGV in real time to maximize the speed of the UGV while driving. Figure 1 presents an example of local path planning with traversability estimation. The algorithm adjusts the path of the UGV to avoid the snowdrift on the front of the platform.

Research on this type of autonomy system at the Łukasiewicz Research Network—Industrial Research Institute for Automation and Measurements PIAP has been conducted since the launch of the project in 2018 called ATENA—Autonomous system for terrain UGV platforms with the following leader function, implemented in the field of scientific research and development work for the defense and security of the state financed by the National Center for Research and Development in Poland under the program Future technologies for defense—a competition of young scientists. The off-road autonomy system was developed and adapted to work with an off-road car. As part of the project, two technology demonstrators were created, functional in following the leader based on the operation of the vision systems. Łukasiewicz—PIAP ATENA system demonstrator has

been tested under real conditions in various weather conditions and is fully operative in the following leader function in rough terrain with obstacle avoidance up to 30 km/h. The main advantage of this system is the fact that the vehicle that is following the leader creates the traversability estimation in real time and calculates its own path to follow the leader, in other words, the following vehicle, not just repeating the leader path; it calculates its own adjustment to the set off-road ability. The developed system is equipped with a module for autonomous navigation in an unknown environment. We paid special attention to solving the issue of map construction and planning a collision-free path.

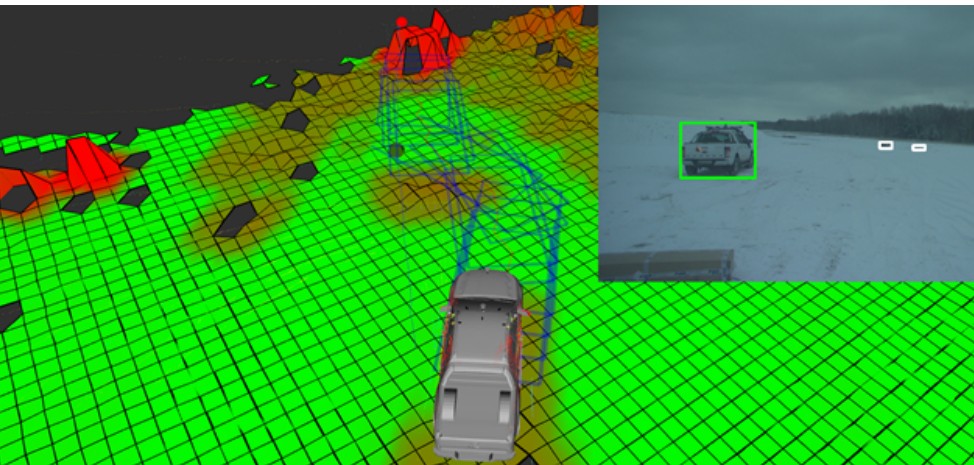

**Figure 1.** Visualization of path planning by a UGV in rough terrain in winter conditions. On the right corner is the image from the roof camera on the UGV.

Figure 2 presents the Łukasiewicz—PIAP ATENA system demonstrators on which the research was performed. The demonstration of the ATENA system contains the off-road car with the Łukasiewicz—PIAP drive-by-wire system and Łukasiewicz—PIAP autonomy controller. The drive-by-wire system allows control of the steering wheel, adjusts the velocity of the vehicle, and controls the brakes, without losing the ability to control the vehicle by a human.

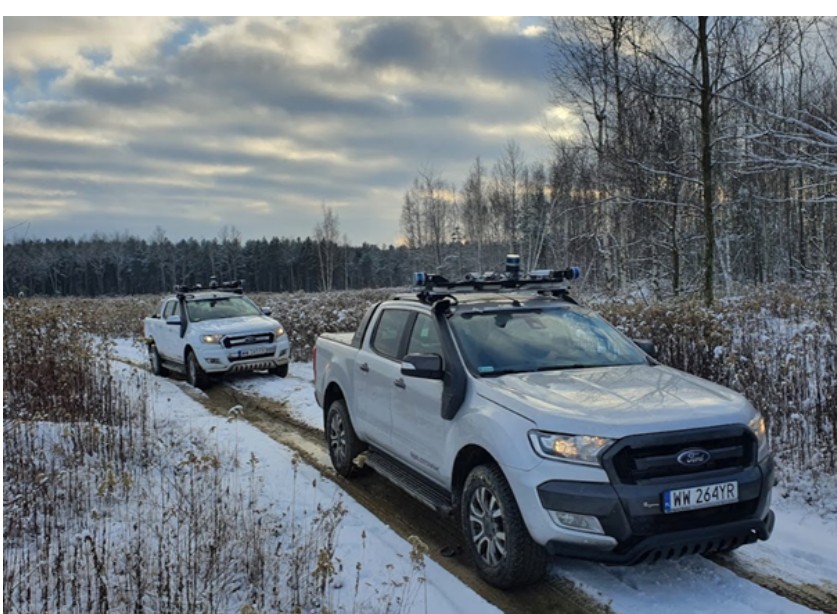

**Figure 2.** Two Łukasiewicz—PIAP ATENA system demonstrators on the test terrain in winter conditions.

This article, which is an extended version of our conference paper [1], presents a method of path planning and building a map of the environment using hexagonal grids. We applied our approach to 3D data obtained using a set of sensors mounted on the ATENA system demonstrator. The data were collected on the premises of the Łukasiewicz Research Network—Industrial Research Institute for Automation and Measurements PIAP. We demonstrate that our algorithm allows us to build accurate models of the environment. The area of the collected data is a rectangle with a circumference of approximately 400 m (Figure 3). The created map is useful for path planning especially in unstructured and rough terrain when the ATENA system demonstrator must search for the lost leader in near historical localization of it or the case of a human leader when the traveling path must be absolutely different than the leader path. The vehicle must calculate its own path not based on the leader path, but based on leader localization.

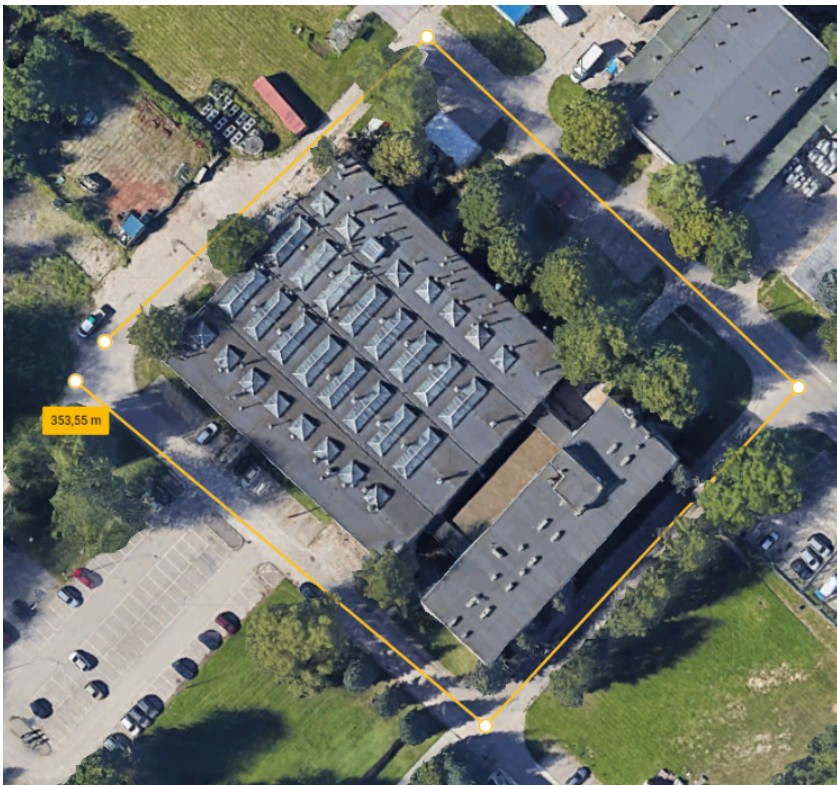

**Figure 3.** The view of the terrain used in this article.

This paper is organized as follows: After the Introduction and the section discussing related work in Section 3, we briefly describe the mapping module. In Section 4, we present a collision-free path planning algorithm. Section 5 contains experimental results illustrating the advantages of our approach. The article concludes with a summary and bibliography.

## 2. Related Work

An essential part of mobile robots is the navigation technique [2]. In [2], it was underlined that the the system has to balance among accuracy, efficiency, and robustness. The navigation consists of three main modules: map building, localization, and collision-free path planning. A map of the environment is necessary for a mobile robot to perform its tasks. When mapping, a robot has to deal with different kinds of noise. Errors are divided into systematic, resulting from defects in the equipment, and nonsystematic, resulting from the conditions of use. Systematic odometry error is caused by a discretized sampling of wheel increments and wheel slippage. Nonsystematic errors result from terrain roughness and wheel slippage. Measurement noise problems also occur with sensors such as laser rangefinders, cameras, etc. Rapid and accurate techniques of data collection, calibration,

and processing are required to improve the accuracy [2]. In the literature, the map building problem is described as a chicken-and-egg problem. The task is to build an accurate map of the environment based on the robot's position and determine the robot's pose based on the created map and the sensors' indication. The reduced accuracy of the map and robot pose have a negative influence on the execution of the path planning task. There are two kinds of navigation systems: reactive navigation and map-based navigation. In the first method, the mobile robot has no map and acts based on the senors' indication [3]. In the case of the map-based navigation, the robot is able to sense, plan, and act. It plans [3] an obstacle-free path to a predetermined destination. Usually, it is assumed that the planned path is optimal, e.g., shortest and fastest. The method consists of four steps:

- The robot observes the environment using sensors;
- Noise is removed;
- The robot determines its pose, and the map is updated;
- A collision-free path is planned.

The first three steps are performed in a loop. The path is replanned if previously undetected obstacles appear on the robot's path.

The choice of the path planning method is closely related to environment representation. The maps described in the literature are divided into two main groups: metric maps and nonmetric maps (topological and semantic). Metric maps are represented as a grid of cells [4] or as a set of features [5].

The grid-based map initially proposed in [4] is one of the primary methods of an environment representation. In this approach, an environment is divided into square areas, and an occupancy value is attached to the corresponding grid cell. Usually, Bayesian theory is used in order to update the occupancy value based on the sensors' reading. Grid-based representation requires an enormous amount of memory, but is able to represent the uniformed objects. The experimental results presented in [6] showed that the improved grid map allows the robot to plan a collision-free path and navigate safely in a static and dynamic environment. A square-based grid map is a popular approach in the ROS system using the Universal Grid Map Library [7]. This library was used in the ATENA system demonstrator with a traversability estimation cost map. The square-based grid map is widely use in path planning based on the Hybrid A* algorithm and uses estimated terrain traversability to find the path that optimizes both traversability and distance for the UGV [8]. A similar approach was tested in winter conditions in the ATENA system demonstrator; the results of the test were sufficient, but it is possible to optimize this approach with a hexagonal grid map. The square cells are also used in rough terrain with high vegetation density when the cells contain information about the "go" and "no go" information [9]. The grid map is also popular because of its ability to represent terrain by the 2.5D grid-type elevation map. Each cell holds a value of the height of the region. This approach can be used even in rough terrain [10]. A hexagonal grid map can be used in mapping static and dynamic environments. The shape of the cell is not important for dynamic approaches, but the advantages of the shape are preserved [11].

Feature-based maps are compact, so they do not require much memory. It is assumed that the features are predefined, so the structure of the environment has to be known in advance.

Square cells used in grid-based representation have some disadvantages. The distance from the center of a cell to the center of a diagonally adjacent cell is greater than the distance to the center of cells with which it shares an edge. Neighboring cells do not always share edges: diagonal cells contact only at a point. Curved shapes are not well represented on a rectangular lattice. In biological vision systems (for example, the human retina), photoreceptors are typically arranged in a hexagonal lattice. It has been shown that hexagonal grids have numerous advantages [12]. First of all, the distance between a given cell and its immediate neighbors is the same along any of the six main directions; curved structures are represented more accurately than by rectangular pixels. A smaller number of hexagonal pixels is required to represent the map. This allows reducing the

computation time and required storage space. In [13,14], it was shown that for a given resolution capability of the sensors, hexagon sampling gives a smaller quantization error. In the paper [15], it was shown that that hexagonal grid map representation was better than the quadrangular grid map representation for cooperative robot exploration, but the problem of collision-free path planning in a real environment was not considered in this article.

In the case of a square grid, we can easily calculate the coordinates of the cell of the array corresponding to the point (x,y). For hexagonal meshes, the calculation is more complex. The lack of an effective method of representing hexagonal meshes and a simple transformation algorithm from Cartesian coordinates to hexagonal coordinates precluded their usage. In [16], the array set addressing (ASA) method was described. The approach is based on representing the hexagonal grid as two rectangular arrays. Figure 4 presents the method of hexagonal grid representation. Different arrays are represented using different colors (pink and blue). The arrays are distinguished using a single binary coordinate. The complete address of a cell in a hexagonal grid is uniquely represented by three coordinates:

$$(a, r, c) \in \{0, 1\} \times \mathbb{Z} \times \mathbb{Z} \tag{1}$$

where $a$—binary index of an array, $r$—row index, $c$—column number, and $\mathbb{Z}$—positive integers. The transformation from hexagonal $(a, r, c)$ representation to Cartesian is defined by the formula:

$$\begin{aligned} x &= d \cdot \tfrac{a}{2} + c \\ y &= d\sqrt{3} \cdot \left(\tfrac{a}{2} + r\right) \end{aligned} \tag{2}$$

where $d$ is the distance between the center of gravity and a vertex of the hexagon.

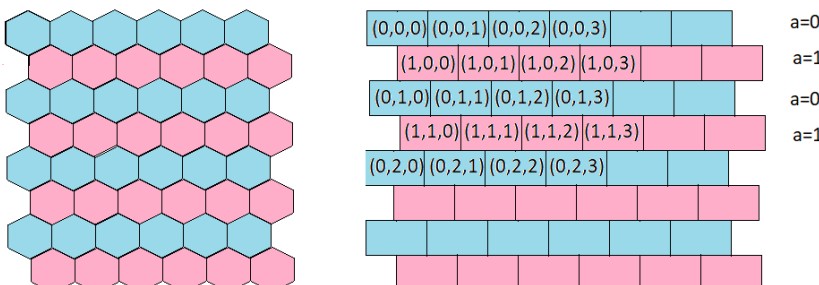

**Figure 4.** Array set addressing method.

An efficient method of transformation from Cartesian coordinates to hexagonal representation was presented in [16].

Figure 5 presents the addresses of the nearest neighbors of the cell $(a, r, c)$ In this article, the hexagonal grid-based representation of the robot environment is presented. It is shown that hexagonal maps of an environment can be easily built based on sensors readings and is useful in collision-free path planning and mobile robot localization.

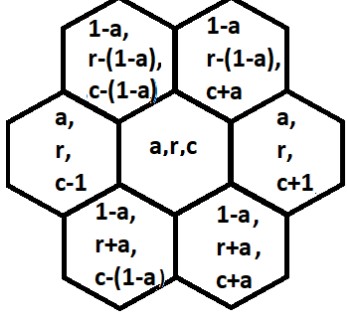

**Figure 5.** The addresses of the nearest neighbors of the cell $(a, r, c)$.

## 3. Mapping

### 3.1. Sensors

The ATENA vehicle is equipped with a set of sensors located on the roof of the off-road car. The sensor set consists of three LiDAR sensors—Velodyne VLP-16 (sixteen layers, 100 m range), five Basler acA1920-48gc (50 fps at 2.3 MP resolution) cameras, and the Xsens MTi-G-710 IMU sensor.

LiDAR sensors are located on the roof of the vehicle in such a way as to create a synergy effect using all three sensors (Figure 6). The main central sensor is tilted by a dozen or so percent so that the cloud of points covers the shape of the road just in front of the ATENA vehicle. Two additional LiDAR sensors are placed on the roof divergently on the sides to form vertical lines of points, which enables better detection of horizontal objects such as barriers, fences, etc. The IMU sensor is placed in the geometric center of the vehicle.

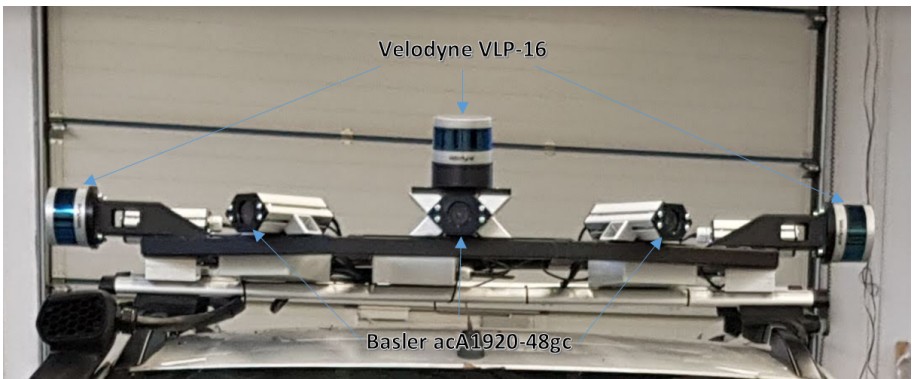

**Figure 6.** Sensors on the roof of the vehicle with the ATENA system.

The present research data were collected using the ATENA core operating with the ROS package. One of the core functionalities of the ATENA program is the fusion of data from Velodyne LiDARs, information from the Xsens MTi-G-710 IMU sensor, and the odometry data from the CAN network of the vehicle. The system builds a 3D world model around the vehicle and creates a map as it moves. For the ATENA system, the model is built in a closed area limited to a square of 50 m × 50 m. To prepare a map of the surroundings, this limitation was turned off, which allowed building a full map of the surroundings around the building.

This work used the resulting point cloud (Figure 7). The point cloud was created by fusing data from the three LiDAR sensors and data from odometry (measurement on the wheels by the onboard system of a vehicle and information from the IMU sensor).

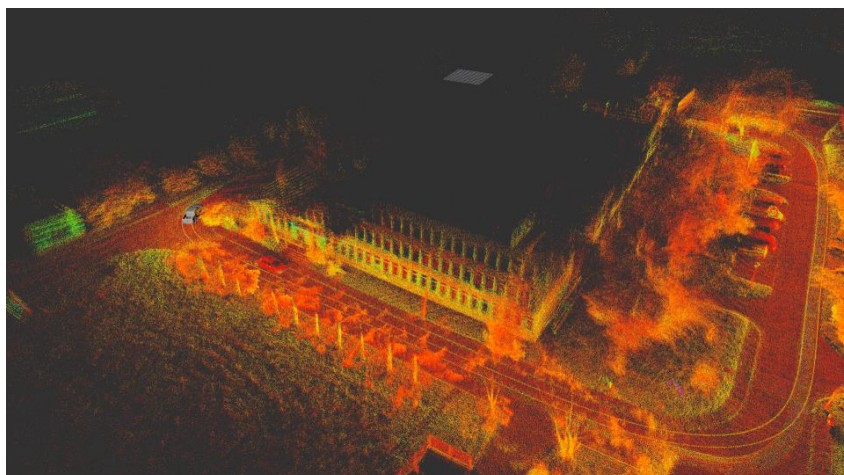

**Figure 7.** Point cloud built by the Łukasiewicz—PIAP ATENA demonstrator.

### 3.2. Map Building

A point cloud can be useful for environment visualization, object detection or feature determination, and semantic segmentation [17,18], but it is not suitable for path planning. Therefore, there is a need to process it into a map. In our approach, a grid-based traversability [19] map was chosen. This is a very popular representation of the environment, and it is also very convenient for path planning. It can be used for algorithms such as A* [20], the diffusion method [21], and reinforcement learning path planning [22].

Two types of lattices were used: square and hexagonal. The former was used as a reference, and the latter is a new approach. There are reasons to believe that the latter is better. As shown in [1], straight lines, circles, and polynomials are statistically better represented on a hexagonal grid than on a square grid. This is important because straight lines are very common on maps, as buildings are composed of such lines. A robot's path, due to the need to avoid obstacles, can resemble a polynomial graph.

In the study described in [1], the quality of the representation of different curves (lines, circles, and polynomials) for square and hexagonal grids was compared. For lines, this was performed as follows. First, the parameters of the straight line were randomly generated. Then, it was drawn on both types of grids (with the same resolution). After this, the average distance between each cell belonging to the representation and the original straight line was calculated. This experiment was repeated 10,000 times, and the average error for all these lines was calculated. The quality of circles and polynomials was checked analogously. Summary results are presented in Table 1. As one can see, for each type of curve, the hexagonal grid represented it better.

**Table 1.** Mean error of the representation of different curves on the square and hexagonal lattice.

| Curve Type | Mean Error for Square Grid | Mean Error for Hexagonal Grid |
|---|---|---|
| line | 0.136 | 0.110 |
| circle | 0.138 | 0.110 |
| polynomial | 0.138 | 0.110 |

To create a grid-based map from a point cloud, we need discretization. Therefore, when creating a map on a square lattice, each point was assigned to a cell as follows:

$$\begin{aligned} x_s &= \lfloor x/w \rfloor \\ y_s &= \lfloor y/w \rfloor \end{aligned} \tag{3}$$

where:

$x$, $y$—original coordinates in the point cloud;
$x_s$, $y_s$—discrete coordinates in the square grid map;
$w$—width of one cell.

Obviously, several points may be assigned to one cell. The height of this cell is calculated based on the average, according to the formula:

$$z_s = \frac{1}{n}\Sigma_{i=1}^{n} z_i \tag{4}$$

where:

$z_s$—height of the cell in the square grid map;
$n$—number of points assigned to the cell;
$z_i$—original z coordinates for all points, which were assigned to a given cell.

The results can be seen in Figure 8. The scale near the map determines the elevation.

The first step necessary to build a map on a hexagonal grid is to convert the Cartesian coordinates to the coordinates used for hexagons. When creating a map, the so-called cube coordinate system [23,24] (this can be seen in Figure 9) is much more useful than the ASA. Therefore, the former was used when generating the map, and the latter was used when

planning the path (a simple way to switch between these coordinate systems is given at the end of this subsection). The transformation between the Cartesian and cube systems is given by the following formulas:

$$
\begin{aligned}
x_c &= x \\
y_c &= \tfrac{\sqrt{3}}{2}y - \tfrac{1}{2}x \\
\zeta_c &= -y_c - x_c
\end{aligned}
\tag{5}
$$

where:

$x, y$—original Cartesian coordinates in the point cloud;

$x_c, y_c, \zeta_c$—continuous cube-hexagonal coordinates.

The next step is to discretize the variables. The following algorithm was used for this purpose [23]. First, auxiliary variables are calculated analogously to the square grid:

$$
\begin{aligned}
\hat{x}_h &= \lfloor x_c/w \rfloor \\
\hat{y}_h &= \lfloor y_c/w \rfloor \\
\hat{\zeta}_h &= \lfloor \zeta_c/w \rfloor
\end{aligned}
\tag{6}
$$

where $\hat{x}_h, \hat{y}_h, \hat{\zeta}_h$—discretized hexagonal coordinates. If the variables satisfy the following condition $\hat{x}_h + \hat{y}_h + \hat{\zeta}_h = 0$, they become the final discrete coordinates on the hexagonal grid. Otherwise, it is necessary to correct the coordinates based on the following Algorithm 1 ($\{\bullet\}$ denotes the fractional part).

---

**Algorithm 1** Discretization of hexagonal coordinates.

---

**if** $\hat{x}_h + \hat{y}_h + \hat{\zeta}_h = 0$ **then**
  $x_h \leftarrow \hat{x}_h$
  $y_h \leftarrow \hat{y}_h$
  $\zeta_h \leftarrow \hat{\zeta}_h$
**else**
  **if** $\{x_c/w\} \geq \{y_c/w\}$ **and** $\{x_c/w\} \geq \{\zeta_c/w\}$ **then**
    $x_h \leftarrow -\hat{y}_h - \hat{z}_h$
    $y_h \leftarrow \hat{y}_h$
    $\zeta_h \leftarrow \hat{\zeta}_h$
  **else if** $\{y_c/w\} \geq \{x_c/w\}$ **and** $\{y_c/w\} \geq \{\zeta_c/w\}$ **then**
    $x_h \leftarrow \hat{x}_h$
    $y_h \leftarrow -\hat{x}_h - \hat{\zeta}_h$
    $\zeta_h \leftarrow \hat{\zeta}_h$
  **else**
    $x_h \leftarrow \hat{x}_h$
    $y_h \leftarrow \hat{y}_h$
    $\zeta_h \leftarrow -\hat{x}_c - \hat{y}_h$
  **end if**
**end if**

---

The height is calculated analogously as before. The results can be seen in Figure 10. In Figure 11, a comparison between the square and hexagonal map is shown. The image shows a close-up of the same southern section of the map, so that the differences between the lattices can be seen more clearly.

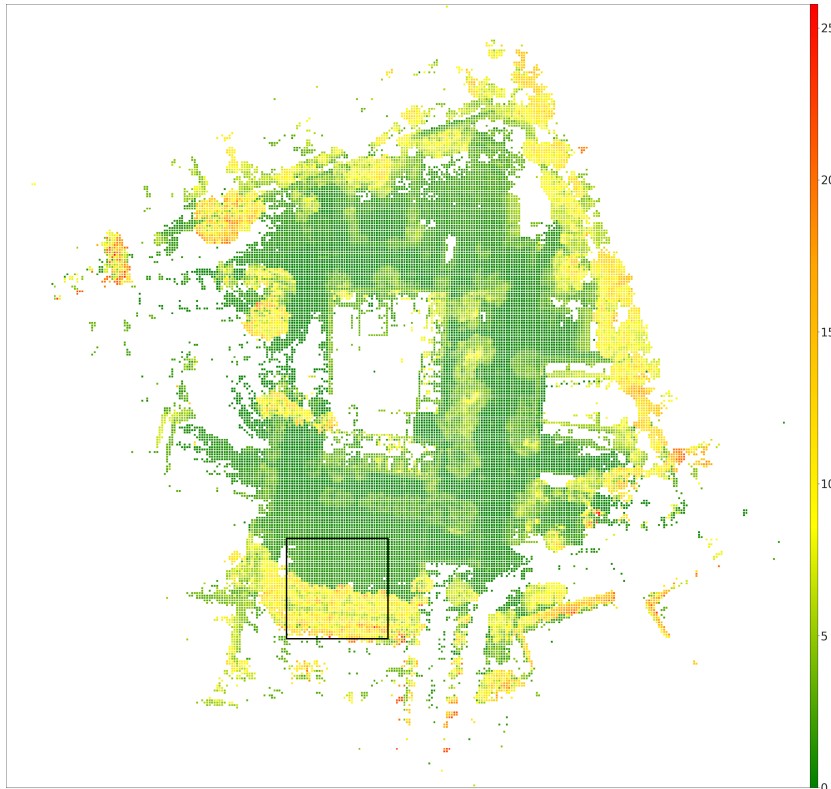

**Figure 8.** The 2.5D map on a square grid. The color indicates at what height an obstacle has been detected within the cell. The scale is placed on the right side of the figure. The black rectangle marks the section shown in the enlarged version in Figure 11.

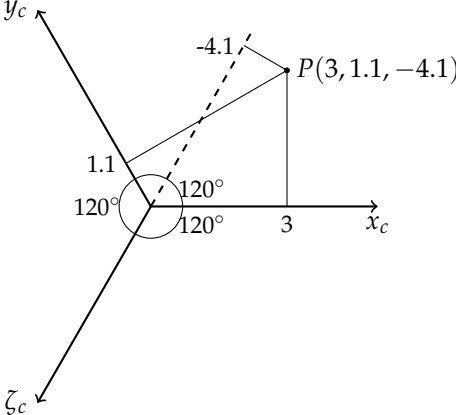

**Figure 9.** Cube coordinate system with an example point.

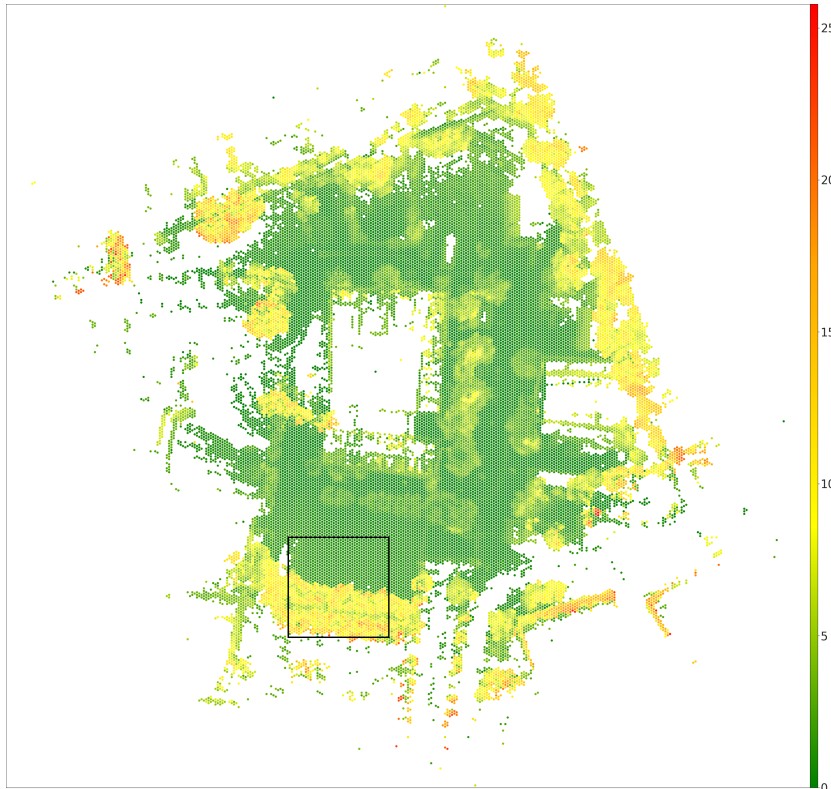

**Figure 10.** The 2.5D map on a hexagonal grid. The color indicates at what height an obstacle has been detected within the cell. The scale is placed on the right side of the figure. The black rectangle marks the section shown in the enlarged version in Figure 11.

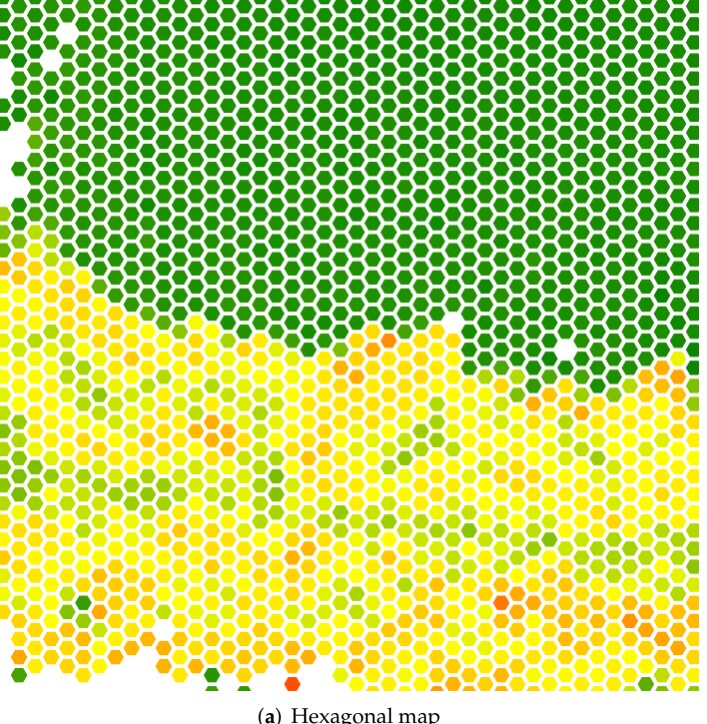

(**a**) Hexagonal map

**Figure 11.** *Cont.*

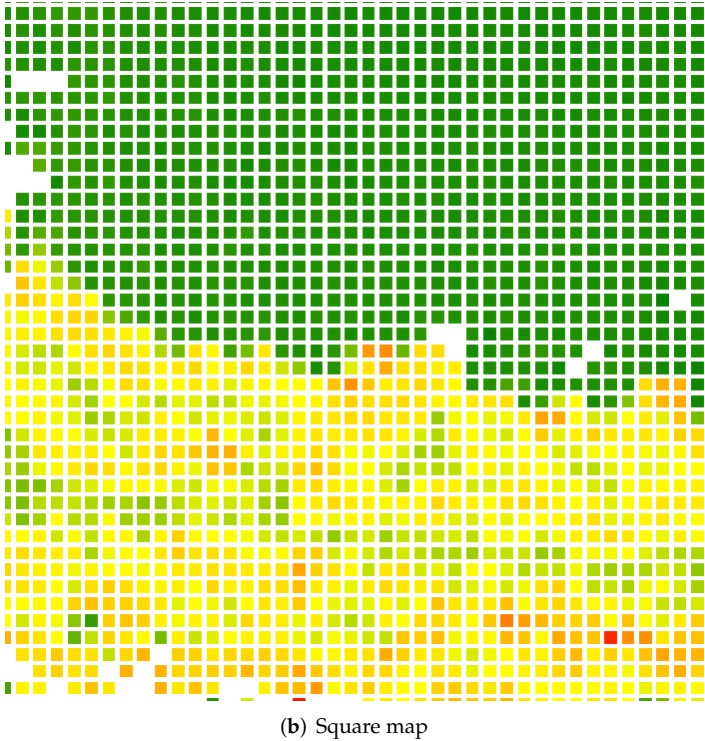

(**b**) Square map

**Figure 11.** Close-up comparison of the grids.

The last step is a conversion between the cube coordinates and ASA according to the formula:

$$\begin{aligned}
a &= x_h \mod 2 \\
r &= \lfloor x_h/2 \rfloor \\
c &= -\zeta_h - \lfloor (x_h + 1)/2 \rfloor
\end{aligned} \tag{7}$$

## 4. Path Planning

In our conference paper [1], we showed that the hexagonal map is useful during path planning. Figure 12 presents collision-free paths computed based on hexagonal and square grids.

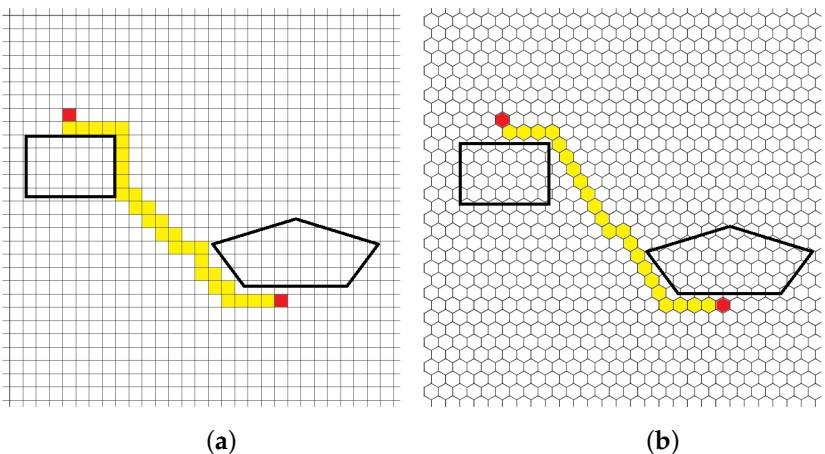

(**a**)                                                    (**b**)

**Figure 12.** Path planning using square grids (**a**) and hexagonal grids (**b**). Black lines represent fragments of obstacles; yellow cells represent the planned path [1].

In this article, we suggest using the diffusion method for path planning [21]. The method is not effective for large areas with few obstacles. For small areas with a large

number of obstacles, the path generation time is shorter than for other methods (potential field, RRT). An additional advantage of the approach is that we can easily take into account the cost of driving over different surfaces. Therefore, we decided to use the diffusion method. In this approach, collision-free path planning is performed on a hexagonal grid. Obstacle-free cells represent the possible robot positions (states). Two states are distinguished: the robot position ($c_R$) and the goal position ($c_G$).

In the first step, a diffusion map is initialized. A big value is assigned to the cell, which represents the goal position ($c_G$), and the values 0.0 are attached to other cells.

In classical path planning systems, we divide cells into two classes: free from obstacles and occupied. For 2.5D maps, class membership is determined by thresholding. A cell is occupied if the observed height exceeds a certain threshold, and free of obstacles otherwise. In classical systems, we look for the shortest path, but we look for the path with the shortest travel time in many cases. The travel time (robot speed) can be related to the type of ground, distance to obstacles, etc. To solve this problem, we introduced an additional parameter cf. In the current version of the system, the value of this parameter is zero for free cells and infinity (very large integer) for occupied cells. In future works, we want to build a surface recognition system and then the parameter will be fully used. In the case of a perfectly smooth surface (asphalt), the value of this parameter equals 0.0. In the case of uneven terrain, the value is increased proportionally to the increase in the cost (time) of movement.

For each unoccupied cell ($c_{aij}$), the value ($v_{aij}$) is calculated according to the formula:

$$v_{aij} = max_{c_{ekl} \in N_{aij}}(v_{ekl} - cf_{ekl} - dist(c_{ekl}, c_{aij})) \tag{8}$$

where:

$cf_{ekl}$—the value of the cost function assigned to the cell $c_{ekl}$, $0 \le c_{ekl} \le \infty$;

$N_{aij}$—neighborhood of the cell $c_{aij}$.

This process continues until stability is established.

During the next step, the list of cells is generated. The first cell represents the robot position. The next one is indicated by the neighbor of ($c_R$) with a maximum value of $v_{akl}$. The process continues until the cell with the maximum function value v is reached.

The method has several advantages:

- The situation in which the position of the robot or the target is surrounded by obstacles is easily detected;
- By specifying the values of the $cf$ function, we can influence the properties of the path—e.g., the distance from obstacles or the type of surface can be taken into account;
- A square cell has four neighbors, and a hexagonal cell has six adjacent cells;
- A path can be smoothed more easily than with a rectangular grid.

## 5. Experimental Results

Experiments were conducted in a real static environment. The path planning method has been tested for a hexagonal map of the environment based on the robot's sensor data. Figures 13–16 show paths generated for several different situations.

R denotes the location of the robot. G denotes the location of the target. Red color indicates cells occupied by obstacles; green color indicates cells free from obstacles; white color indicates cells whose state is unknown; black cells represent the path. Figure 13 shows a situation in which the robot avoids moving through unknown areas (cf = 100), but the distance to obstacles has not been taken into account. It is clear that the path runs dangerously close to the obstacles.

In Figure 14, an additional cost was added to the cells directly adjacent to obstacles.

In Figure 15, the neighborhood radius was extended by three cells.

For different values of the radius of interaction of the obstacles, we obtain different paths. The path shown in Figure 15 is longer than the path in Figure 13, but it is safer.

Figure 16 shows a situation where the cost function is the same for free and unknown cells and the radius of interaction of obstacles is three cells. The path is completely different from that shown in the previous figures.

We performed three series of experiments to compare the lengths of the path generated by the diffusion method in the case of hexagonal and rectangular grids. In each series, 1000 start and destination points were generated. Paths were planned using the methods mentioned above. The generated lists of cells were converted into lists of segments, and then, the lengths of the paths were calculated.

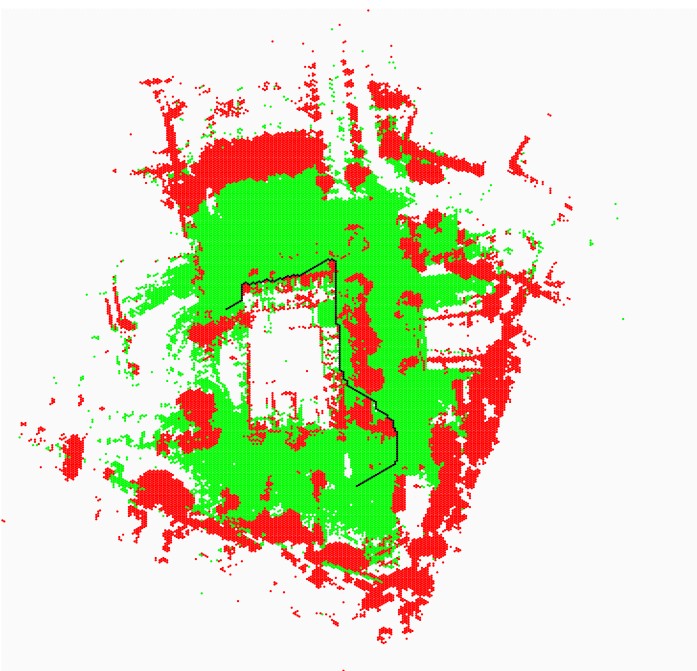

**Figure 13.** Collision free path—classic approach, green for obstacle-free cells, red for obstacle-occupied cells, and black for the planned path.

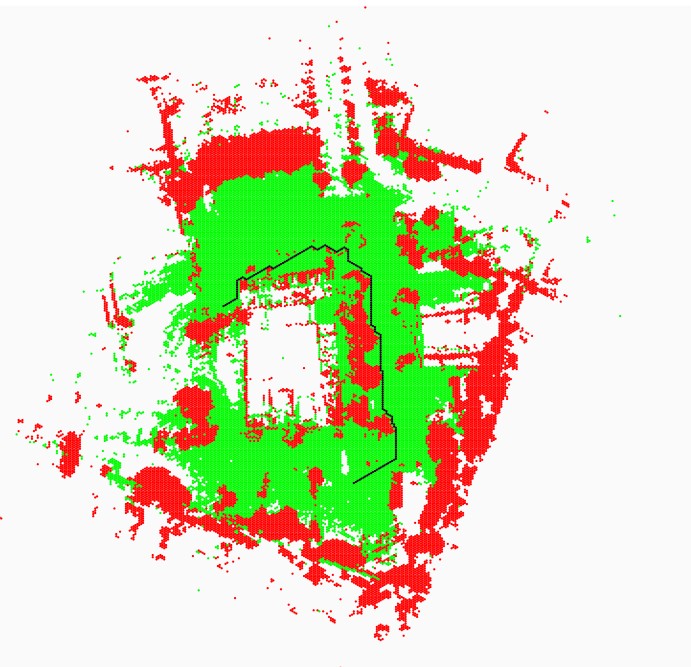

**Figure 14.** Collision free path—distance to the obstacles is taken into account (radius of neighborhood = 1), green for obstacle-free cells, red for obstacle-occupied cells, and black for the planned path.

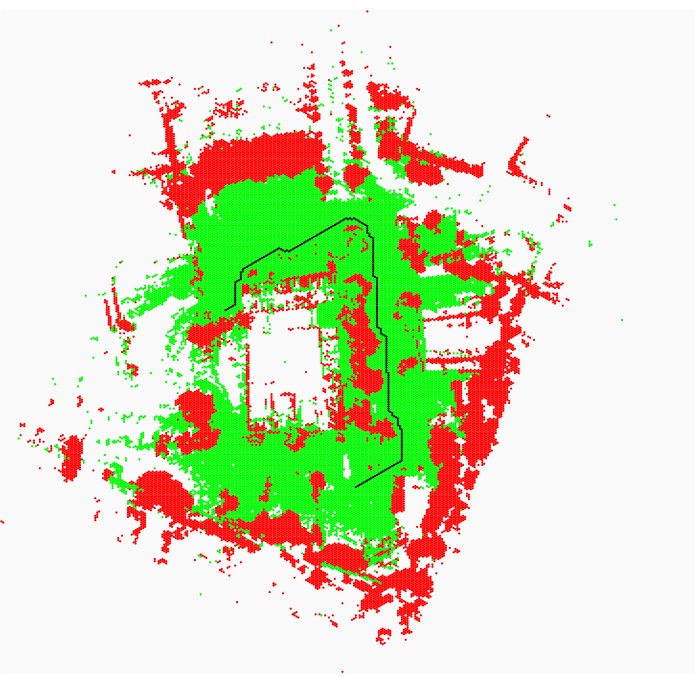

**Figure 15.** Collision free path—distance to the obstacles is taken into account (radius of neighborhood = 3), green for obstacle-free cells, red for obstacle-occupied cells, and black for the planned path.

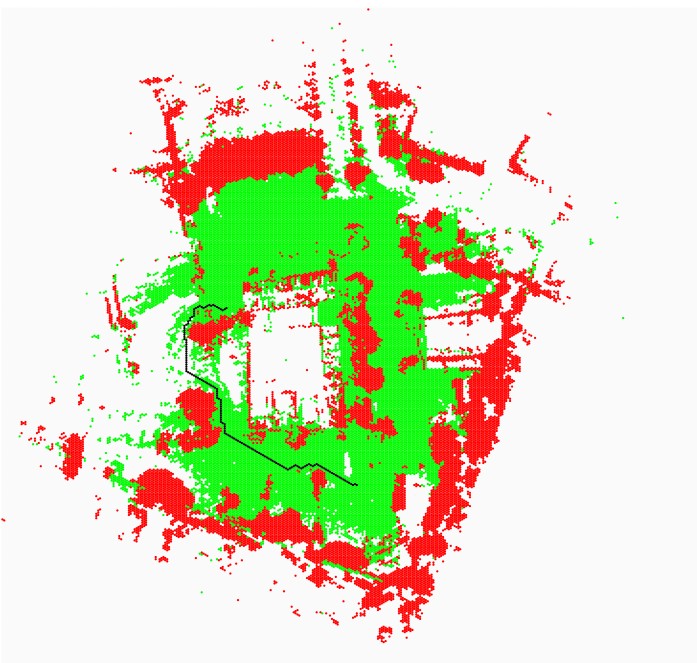

**Figure 16.** Collision free path—the cost function is the same for free and unknown cells, green for obstacle-free cells, red for obstacle-occupied cells, and black for the planned path.

The value of parameter dd is computed as follows:

$$dd^i = \frac{(dr^i - dh^i)}{dh^i} \tag{9}$$

where: $i$—the number of experiments; $dr$—the length of a path generated using square grids; $dh$—the length of a path generated using hexagonal grids.

In grid-based path planning methods, we need to expand obstacles by a given number of cells. This operation is equivalent to the dilation used in machine vision. We performed each series of experiments for a different neighborhood value (from r = 0 to r = 2). The results of the experiments are presented in Table 2. Figure 17 shows histograms of the parameter dd and corresponding Gaussian KDE distributions.

It can be seen that when we do not expand the obstacles, the gain from using the hexagonal grids is about 3%, while it exceeds 10% when we expand the obstacles. The shortened path length for the hexagonal grid is due to the fact that the hexagonal grid represents the shape of the obstacles better than the rectangular grid. For r = 2, the graph is not symmetrical with respect to the mean value, and in 82% of the cases, the path planned with the hexagonal grid is shorter than with the rectangular grid.

We did not notice a significant effect of map form (rectangular grid, hexagonal grid) on path planning time. The most time-consuming stage was diffusion. Regardless of the grid type, the number of cells occupied by obstacles affected the diffusion process. The diffusion process took more than one second (computer, map) for an empty environment. Fortunately, the absence of barriers is easy to detect, and we can plan the path using ordinary geometrical methods. In a maze-type environment with many obstacles, the diffusion time did not exceed 0.5 s (PC, Windows 10, i5-1035G1 CPU 1.00 GHz, 1.19 GHz, RAM 16 GB, map: $640 \times 320$ cells; a cell represents 1 m$^2$ in area). An essential advantage of the hexagonal mesh is that it approximates the shape of objects much better (Section 3). As a result, in many situations, a collision-free path is not found when using a rectangular mesh, but is found when using hexagonal one. As a result, the generated path transforms to the interpretable data by the Łukasiewicz—PIAP drive-by-wire system, which works on the universal automotive-grade controller for complex mobile working machines. The low-level program was designed to control the steering wheel by the original power steering of the off-road car. The velocity of the vehicle is controlled by the algorithm by using the electronic throttle and added electric ABS pomp. The universal controller is responsible for adjusting the velocity of the vehicle and the steering angle of the wheels to the set value by the CAN frame sent by the Łukasiewicz—PIAP autonomy controller.

**Table 2.** Mean values of parameter dd, for different neighborhood values.

| Neighborhood Value | Mean dd |
|---|---|
| 0 | 0.03 |
| 1 | 0.12 |
| 2 | 0.15 |

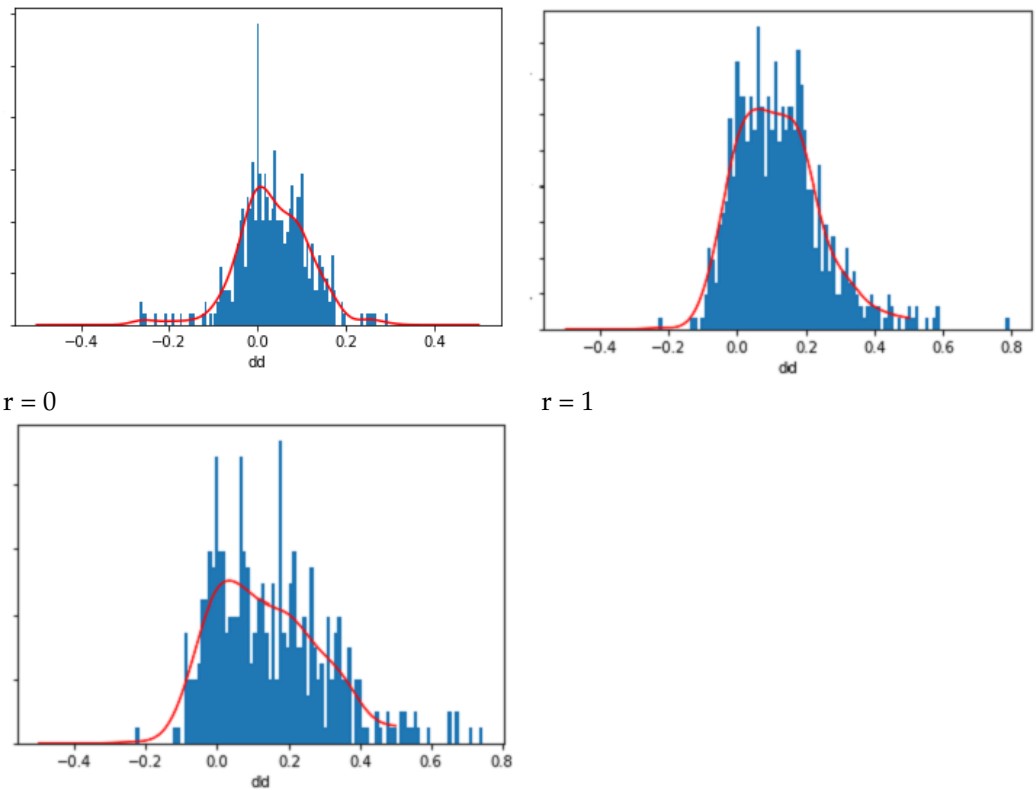

r = 0        r = 1

r = 2

**Figure 17.** Histograms of parameter dd; the red line represents the estimated probability density function.

## 6. Conclusions and Future Works

In this paper, we presented the possibility of using hexagonal grids in outdoor navigation. Experimental results showed the advantages of a hexagonal grid over a square grid. In our further work, we will use the diffusion map to determine safe vehicle speeds. Furthermore, we plan to determine robot localization based on matching hexagonal maps from two places.

The method could be used in the autonomy systems for the outdoor navigation in the UGV control systems in convoys, reconnaissance, surveillance missions, etc. This type of mapping can bring the goals of speedup and improved local path planning in real time on autonomous systems to reach the level of not limiting the UGV ability on rough terrains. We also plan to apply elements of the presented algorithm to the tasks described in the papers [25,26].

**Author Contributions:** Methodology and writing: B.S., P.D. and R.W.; software for path planning, B.S.; software for mapping, P.D.; low-level control, R.W. All authors have read and agreed to the published version of the manuscript.

**Funding:** This research received no external funding.

**Institutional Review Board Statement:** Not applicable.

**Informed Consent Statement:** Not applicable.

**Data Availability Statement:** The source code and data used to support the findings of this study are available from the corresponding author upon request.

**Conflicts of Interest:** The authors declare no conflict of interest.

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
