# Peer review of "Hexagonal Grid-Based Framework for Mobile Robot Navigation"

_remotesensing, doi:10.3390/rs13214216_

Round 1
Reviewer 1 Report
The authors have taken into account most of previous remarks.
The introduction have been a bit extended to strengthen the link between the following-car project and the rest of the paper. The authors emphasize the fact that the task realized is not a strictly following since the second car do not follow same path of the first, but it could be expressed as a “moving goal” rallying task.
The bibliography have been extended by adding several papers using square-cells grids for map construction and path planning. The authors should also add the two following references concerning previous works on hexagonal grids. [1] Luczak and Rosenfeld, "Distance on a Hexagonal Grid," in IEEE Transactions on Computers, vol. C-25, no. 5, pp. 532-533, May 1976, doi: 10.1109/TC.1976.1674642. [2] Herrmann, Daniel & Kamphans, Tom & Langetepe, Elmar. (2010). Exploring Simple Triangular and Hexagonal Grid Polygons Online.
In section 3 , some notation have been changed and some detail have been added about the coordinate conversion making this part more complete and understandable. Figure 11, was added as a zoomed view of figures 8 and 9. The zooming level is sufficient, but the author should relocated these figures to put them closer to the full map. They also could add a rectangle showing which part of the map is zoomed. The authors should also indicate the resolution used for the grid.
In section 4, the authors have precised they used a threshold convert the 2.5D map to occupancy, this imply to have a very accurate scan alignment following Z axis (in particular tilt and roll compensation) unless the threshold may miss-classify road and obstacle. An explanation about the parameter c have been added but it need some additional details, its seem related to the traversability but it is not clear.
Finally, a Monte-Carlo evaluation have been added to support the proposition of the authors. This evaluation is quite minimal but sufficient to quantify the reduction (between 3% and 10%) for a length of a path planed into an hexagonal grid compare to a path plan into a square-grid.
Typos: L268: dr => dh
Reviewer 2 Report
This paper presents a novel method of occupancy grid mapping using hexagonal grids. Traditional occupancy grids are square which makes the diagonal cells have different center-center distance than the other neighboring cells. This disadvantage is not present in the hexagonal grids where all the neighboring cells are equidistant. The authors present some results for the lesser amount of error in hexagonal system as opposed to the square occupancy grid.
The paper is written well and it deals with an important concept. However, a major concern I have is that this concept has been explored previously in Quijano, H. J., & Garrido, L. (2007, September). "Improving cooperative robot exploration using an hexagonal world representation." The authors have not provided this reference and how their paper distinguishes itself from the previous reference. Another concern is the lack of demonstrable results with this technique.
Author Response
References to the article (Quijano, H. J., & Garrido) have been added, and we have described the differences between our and the cited article.
Reviewer 3 Report
This paper investigates the problem of mobile robots navigation. The topic is interesting, nevertheless the manuscript should be further improved. Here are some issues which may improve the quality of the paper.
--Relevant work is comparatively weak to support the points of this paper. More recent works should be discussed, especially in the past three years, such as Effective Safety Strategy for Mobile Robots Based on Laser-Visual Fusion in Home Environments, and the articles published in Special Issue on Mobile Robots Navigation on Applied Sciences (MDPI).
--For hexagonal grid map generation, an off-line job is implemented, which will bring some gaps to the practical application of outdoor robots. Besides, how to deal with obstacles in space of 3D point cloud representation? The explanation needs to be provided.
-- For the practical applications of robots, the time cost is a very important factor. In the experiments of this paper, the planning time cost is not given to demonstrate effectiveness.
--More experiments should be carried out, including dynamic scenarios and the navigation experiments in the real world.
Reviewer 4 Report
This paper presents the mobile robot navigation using hexagonal lattice. The authors claim that hexagonal maps provide some advantages in path planning compared to traditional rectangular grid maps. To verify this claim, a small-scale experiment using Łukasiewicz - PIAP ATENA system has been performed.
The work seems promising as it tries to explore the benefits of the hexagonal grid maps for navigation. However, I am disappointed with the lack of proper discussions of the experimental results and the overall presentation of the paper, which should be improved significantly. Besides this, the following issues need to be fixed.
- The paper contains several typos and grammatical errors. Please fix them.
- In line 87, you would like to make clear about noise.
- You would like to put citations before full stops, which is more prominent in the paragraph beginning with line number 96.
- Please explain notations in equations adequately. In (1), probably 'Z' refers to Integer. Also, in (2), the notation of 'd' is missing. Also, make clear what n in (4) means, and the subscripts h and c in (5)-(7) points.
- You would like to make the statements in lines 226-228 more clear. Perhaps, some equation.
- The captions of the figures should be improved. They should contain all relevant information. e.g., authors can add what different color components in the figures mean.
Round 2
Reviewer 2 Report
Thanks for the changes to the manuscript. In the results, for Figure 17, you show the histograms and provide the mean values in Table 2. However, my question is whether the mean is representative when the plot is skewed (seems like the red line in the figure is the fitting but its not mentioned in the figure). It seems that Table 2 should be revised to provide additional data of distributions.
Reviewer 3 Report
I appreciate the author's efforts in revising the paper. But the response was not entirely satisfactory. Why only static scenarios experiments were designed. For the real world, dynamic scenarios are more common. Therefore, the validation in actual dynamic scenarios can better reflect the applicability of the method.Author Response
Please see the attachment.

Reviewer 4 Report
The authors have answered my concerns, some of them not very satisfactorily but in an acceptable way. I still have the feeling that the overall presentation of the paper can be improved. The following comments should be considered in the revised manuscript.
- You would like to specify the types of noises (line number 92).
- You would like to represent an integer symbol in equation (1) by \mathbb{Z} instead of Z.
- You would like to improve the flow of writing in Section 1: Introduction.
- You would like to correct typos. e.g.,
Fig. 2 instead of Fig.2 (line number 46), section 3 instead of sec. 3 (line number 67),
add full stop after the first sentence in the caption of Fig. 8,
and so on. - You would like to define what notation of 'd' means in equation (2).
- You would like to make References uniform. e.g.,
[1] can be kept as a footnote or use the proper citation style for web pages,
in [3] Applied Sciences instead of Appl. Sci. for consistency,
and so on.
Author Response
Please see the attachment.

This manuscript is a resubmission of an earlier submission. The following is a list of the peer review reports and author responses from that submission.
Round 1
Reviewer 1 Report
This paper present a navigation system for autonomous vehicle. More precisely, it presents a mapping process that create an hexagonal-cells occupancy-grid from lidar data and it also presents the use of this map for path planning. This paper is presented as an extended paper of [1]. Because I do not have access to this conference paper, I can not evaluate what is the novelty compared with previous paper.
The introduction presents the project and the platform that support the work presented but it seem out of topics because the case addressed in the rest of the paper do not deal with task "vehicule following" and "off-road" context. The related works section is quite short on mappings approaches (only 3 papers) and presents majors previous works on hexagonal-grids but misses some important ones [2, 3]. The Mapping section present the hexagonal-grid map construction algorithm that is basically a transformation between points of world 2D plane (2D Cartesian) to hexagonal-grid indexes (3D). The section is not very clear, firstly the authors use $z$ for the height coordinate but $z_c$ and $z_h$ for third hexagonal indexing dimension. Secondly, the equation following eq. 6 is not numbered and I do not understand what the authors want to explain here. The method produce an well known 2.5D grid where each cell store the mean height of the points insides. Results of grids constructed from a point-cloud are showed (fig. 8 and fig. 9) to illustrate the improvement of hexagonal-grids but pictures are too small, and need to be enlarge at least with factor 4. The part Path Planning present the algorithm used to compute the trajectory using the previous grid. A basic diffusion algorithm is used here. The authors do not precise how the convert 2.5D grid to an binary occupancy grid (by applying threshold ?). The cost $c_{aij}$ is not clearly introduced, how this parameters is fixed ? The subscript in $v_{aij}$ is confusing because $i$ and $j$ are dummy variables whereas $a$ is already the name of the variable in “arc” indexing. Experimental results section present the qualitative result of path planning using the hexagonal-grid built in section 3 and the algorithm presented in section 4. The results consist in illustrations that represent the grid and the path predicted on only one configuration (start/end) using 4 planning setups (obstacles distances and unknown space allowed or not). Finally, the conclusion resumes the main contributions of the paper and presents the next works.
This paper some presents majors issues. The first issue is that even if the authors argue hexagonal-grid is better for path planning than squared-grid they do not prove it at all. There is no comparison under any criteria between the 2 representations. Fig.11 showing a shorter path could easily be contested if we consider a 8-neighborhood for the square grid or if we apply a rotation of 45°. An other issue is that I do net see what the authors want to show and what is the contribution of the planning path section. The algorithm presented is not evaluated and not compared to any other approach. The conclusions of the discussion are quite logical, if we add a weight to be close of obstacles, the path become longer but further of obstacles. Moreover, the diffusion algorythme seem not the best option here, it is very slow to update because need to compute the value of all the cells in the grid, that is not very compatble with a real time system. To solve these issues, the authors should provide an extensive evaluation comparing with quantitative metrics on a large set of random configurations (start/end) the path planned using regular grid and hexagonal-grid and show the improvement using objective metrics.
Finally, considering the previous problems I consider that t the contributions presented in this papers have an overall low impact. But since this paper is an extended version of an already publish conference paper, and because I do not have access to this paper, I can not declare but I wonder what is overlaps between the two and if the contribution of this paper are not even lower.
Reviewer 2 Report
The authors present an alternate hexagonal grid-based framework for discretizing the world instead of the regular square grid used in occupancy mapping framework. The proposed method has some significant advantages - especially in the neighborhood query which is similar for neighboring hexagonal cells in every direction. The authors show that the proposed method can qualitatively provide better results for both mapping and navigation.
The paper is overall very well-written. The only major problem I have is with the absence of quantitative results. How much of a difference are we getting in terms of amount of information/cost function utility using the proposed method instead of the square grid? When is the difference significant? Does the increase in computational complexity justify the benefits?